# ^1^H-Nuclear Magnetic Resonance Analysis of Urine as Diagnostic Tool for Organic Acidemias and Aminoacidopathies

**DOI:** 10.3390/metabo11120891

**Published:** 2021-12-20

**Authors:** Ninna Pulido, Johana M. Guevara-Morales, Alexander Rodriguez-López, Álvaro Pulido, Jhon Díaz, Ru Angelie Edrada-Ebel, Olga Y. Echeverri-Peña

**Affiliations:** 1San Ignacio University Hospital, Bogota 4665684, Colombia; nfpulido@husi.org.co; 2Institute for the Study of Inborn Errors of Metabolism, School of Sciences, Pontificia Universidad Javeriana, Cra 7 # 43-82, Edificio 54 Lab 303A, Bogota 4665684, Colombia; johana.guevara@javeriana.edu.co; 3Molecular Biology and Immunology Department, Fundación Instituto de Inmunología de Colombia (FIDIC), Bogota 4665684, Colombia; rodriguez.edwin@javeriana.edu.co; 4Chemistry Department, School of Sciences, Pontificia Universidad Javeriana, Bogota 4665684, Colombia; diaz.john@javeriana.edu.co; 5Electrical and Electronics Department, Universidad Nacional de Colombia, Bogota 4665684, Colombia; apulidoa@unal.edu.co; 6The Natural Products Metabolomics Group, Strathclyde Institute of Pharmacy and Biomedical Sciences, Faculty of Science, University of Strathclyde, The John Arbuthnott Building, 161 Cathedral Street, Glasgow G4 0RE, UK

**Keywords:** metabolism, Inborn Errors of Metabolism, organic acidemia, aminoacidopathy, nuclear magnetic resonance

## Abstract

The utility of low-resolution ^1^H-NMR analysis for the identification of biomarkers provided evidence for rapid biochemical diagnoses of organic acidemia and aminoacidopathy. ^1^H-NMR, with a sensitivity expected for a field strength of 400 MHz at 64 scans was used to establish the metabolomic urine sample profiles of an infant population diagnosed with small molecule Inborn Errors of Metabolism (smIEM) compared to unaffected individuals. A qualitative differentiation of the ^1^H-NMR spectral profiles of urine samples obtained from individuals affected by different organic acidemias and aminoacidopathies was achieved in combination with GC–MS. The smIEM disorders investigated in this study included phenylalanine metabolism; isovaleric, propionic, 3-methylglutaconicm and glutaric type I acidemia; and deficiencies in medium chain acyl-coenzyme and holocarboxylase synthase. The observed metabolites were comparable and similar to those reported in the literature, as well as to those detected with higher-resolution NMR. In this study, diagnostic marker metabolites were identified for the smIEM disorders. In some cases, changes in metabolite profiles differentiated post-treatments and follow-ups while allowing for the establishment of different clinical states of a biochemical disorder. In addition, for the first time, a ^1^H-NMR-based biomarker profile was established for holocarboxylase synthase deficiency spectrum.

## 1. Introduction

Inborn Errors of Metabolism (IEM) are monogenic diseases that affect the normal functioning of the human metabolism due to mutations in enzymes, transporters, and co-enzymes, among other proteins directly or indirectly involved in a metabolic pathway. IEM classification could be based on either the involved metabolic pathways of amino acids, fatty acids, carbohydrates, etc., or affected organelles, as in the case of lysosomal storage diseases and peroxisome alterations [1,2,3]. Specifically, small molecule Inborn Errors of Metabolism (smIEM) are disorders in the metabolism of carbohydrates, purines, pyrimidines, creatine, and vitamins, as well as organic acidemias and aminoacidopathies. These groups of IEMs are characterized by clinical indications of intoxication, mainly caused by the abnormal production (usually a result of alternating pathways) of toxic metabolites. Clinical manifestations of smIEM include food intolerance or enteral food rejection, drowsiness, hypotonia, seizures, rapid clinical deterioration leading to coma, and even death within hours [3,4] These manifestations could be unpredictably prompted after a period of complete normality lasting from a few days to several months [3]. In general, a patient’s phenotype has a wide-ranging variability, from moderate and mild indications that can manifest at any age to severe neonatal onset commitments that could have fatal outcomes. Early and accurate diagnosis becomes a medical emergency when considering that the timely treatment of this type of disorder has the possibility to make changes in disease prognosis and favourable impacts in quality of life while avoiding or reducing neurological sequelae. In fact, it is important to consider that effective treatments are available for many smIEM including nutritional intervention, pharmacotherapy, supplements, life style changes and transplants, among others [3,4,5].

Conventionally, the smIEM diagnostic approach for symptomatic patients has been based on the study of the abnormal appearance and/or elevation of specific metabolites in urine or plasma. For instance, amino acid quantification by ion-exchange chromatography (either using high-performance liquid chromatography (HPLC) or liquid chromatography coupled to mass spectrometry (LC–MS/MS)) in the study of aminoacidopathy and urinary organic acid determination by gas chromatography coupled to mass spectrometry (GC–MS) are the gold standard techniques used to establish the diagnosis of organic acidurias [5,6,7].

Recently, technical and conceptual advancement in metabolomics has led to the use of analytical tools such as proton nuclear magnetic resonance (^1^H-NMR) for the global study of metabolites generated in biological organisms due to changes in physiological states, nutritional habits, disease conditions, and treatment with drugs, among others [8,9,10]. A metabolomics approach utilizes multivariate analysis to generate plots that are easy to visualize and interpret, thus saving time and resources while contributing to greater effectiveness in the diagnostic approach [8]. Considering the clinical manifestation overlaps that may occur among different smIEMs, a metabolomics approach allows for the overall and simultaneous evaluation of metabolic pathways associated with this group of diseases, thus enabling the rapid and specific identification of the affected metabolism, which reduces the morbidity and mortality from this type of disorder in the long-term [9,10,11].

Accordingly, various research groups have introduced the use of ^1^H-NMR-based metabolomics using high-resolution equipment (>500 MHz) to obtain promising results; however, the accessibility to such technology might be limited in some contexts [12,13,14,15,16,17]. Therefore, in this work, we aimed to establish a ^1^H-NMR-based metabolomics approach using equipment with a field strength of 400 MHz as an initial screening technique for the urine diagnostic approximation of smIEM in combination with GC–MS. Initially, the average spectrum of the sample population was determined by using 36 samples from healthy volunteers of different ages from 6 days to 14 years old. Simultaneously, the biochemical diagnosis of several smIEMs using the gold standard test of GC–MS was validated. The urine samples of diagnosed patients were then analysed by ^1^H-NMR. The qualitative analysis of the profiles allowed for the identification of the metabolites previously reported in the literature with higher-resolution equipment. The results obtained in this study allowed for the identification of changes in metabolite profiles that differentiated post-treatments and follow-ups, summing to the possibility of establishing metabolic differences among various clinical states of a biochemical disorder. In addition, for the first time, a ^1^H-NMR-based biomarker profile was established for holocarboxylase synthase deficiency, a rare treatable disorder caused by defective carboxylase coupling to biotin as a coenzyme. Finally, quantitative analyses suggested that a ^1^H-NMR-based metabolomics approach at a field strength of 400 MHz allowed for appropriate differentiation between healthy and smIEM individuals.

## 2. Results

^1^H-NMR spectral data and GC–MS chromatographic profiles of all samples were obtained to qualitatively compare the results from both analytical techniques. Different biochemical abnormalities were observed in the analysed samples. Table 1 shows the summary of abnormalities detected for each smIEM disorder studied in this work with both methodologies.

### 2.1. Profile for Healthy Population

The average ^1^H-NMR spectrum for non-affected individuals was established using urine samples from the control group (Figure 1); it was used as the reference data for the qualitative analysis of the spectral data obtained from individuals diagnosed with the occurring IEM disorders. For analysis, ^1^H-NMR peaks in the profile were grouped according to chemical families (Figure 1). In Table 2, the detail regarding metabolites detected in the spectra for the control group are presented with their ^1^H-NMR chemical shifts and peak multiplicities. Figure 2 shows a representative GC–MS profile of the control group. 

### 2.2. smIEM Profiles

For qualitative analysis, ^1^H-NMR spectral data from pathological samples were compared to the average control spectrum to enable the description of significant metabolic changes between healthy and IEM-affected individuals. In general, pathological samples mainly showed changes in the region between 1.0 and 4.2 ppm. In addition, GC–MS chromatographic profiles of all samples were also obtained to qualitatively compare the results from both analytical techniques. In all cases, some differences were found with regard to detecting respective metabolites when comparing the results obtained between the two analytical techniques (Table 1). 

### 2.3. Propionic Acidemia

The ^1^H-NMR spectra (Appendix A) and GC–MS chromatogram (Appendix A) obtained from patients with propionic acidemia exhibited the occurrence of the excretion of the characteristic metabolite profile for propionic acidemia accompanied by ketosis, including propionylglycine, 3-hydroxy-propionic acid, and propionic acid, which are typical for the disorder (Table 1). In fact, the occurrence of propionic acid was only observed by ^1^H-NMR. 

### 2.4. Isovaleric Acidemia

The obtained ^1^H-NMR spectra varied according to the clinical condition of the patient. Regarding the patient’s clinical state, characteristic changes could be observed. In samples obtained from individuals experiencing an acute episode of isovaleric acidemia, the occurrence of 3-hydroxyisovaleric acid (3-OHisoV), isovalerylglycine (Ivg), and isovalerylalanine (Isov-ala) were identified; however, after treatment, the marker metabolites of this disease disappeared. The chemical shifts of these metabolites are listed in Table 1 (Appendix A). The GC–MS chromatograms showed 3-isovalerylglutamate but no isovalerylalanine (Appendix A) (Table 1).

### 2.5. 3-Methylglutaconic Acidemia

The ^1^H-NMR spectra of two samples indicated the occurrence of 3-methylglutaric acid (3-MG). The presence of very weak signals with no defined multiplicity was also found between 1.97 and 2.09 ppm, along with doublet pairs at 3.45 and 3.54 ppm, which may correspond to the methylated protons of 3-methylglutaconic acid (Table 1) (Appendix A). By GC–MS, peaks for 3-MG and 3-methylglutaconic acid (Appendix A) (Table 1) were also found.

### 2.6. Glutaric Acidemia Type I

Signals belonging to glutaric acid were observed in the ^1^H-NMR spectra of the urine samples of patients with glutaric acidemia type I (Table 1) (Appendix A). GC–MS also indicated the occurrence of 3-hydroxyglutaric and glutaconic acids (Appendix A) (Table 1).

### 2.7. Medium Chain Acyl-CoA Dehydrogenase Deficiency

For this deficiency, a complex ^1^H-NMR spectrum indicated the occurrence of several pathological metabolites, including hexanoylglycine, 2-methylbutyrylglycine, isobutyrylglycine, isovalerylglycine, butyrylglycine, suberylglycine, hexanoyl-carnitine. and octanoyl-carnitine (Table 1) (Appendix A). GC–MS demonstrated the presence of adipic acid and phenylacetylglutamine (Appendix A) (Table 1).

### 2.8. Lactic Aciduria

For lactic aciduria, an increased signal intensity for lactic acid was observed in the sample ^1^H-NMR spectrum, accompanied by the occurrence of 2-hydroxy-isovaleric acid, acetic acid, and glucose (Table 1) (Appendix A). GC–MS analysis demonstrated the very high excretion of lactic acid and elevated concentrations of 2-hydroxy-isobutyric acid and 4-hydroxyphenylactic acid but only a slight increase of 2-hydroxy-isovaleric acid (Appendix A).

### 2.9. Maple Syrup Urine Disease (MSUD)

The ^1^H-NMR spectra for MSUD showed changes between 0.70 and 4.22 ppm when compared to the spectra of urine samples from healthy individuals. In different patients, we detected the occurrence of pathological metabolites that included isocaproic acid, alloisoleucine, isoleucine, 3-methyl-2-oxo-valeric acid, and ketoleucine (Appendix A). The GC–MS chromatogram presented a similar pattern (Appendix A) (Table 1). The presence of ketoleucine was only observed by ^1^H-NMR (Table 1).

### 2.10. Phenylalanine Metabolism Disorders

For phenylalanine metabolism disorders, the ^1^H-NMR spectrum exhibited an increase of resonances in the region between 7.20 and 7.42 ppm corresponding to signals of aromatic compounds such as 3-phenyllactic acid and N-acetylphenylalanine (Appendix A). GC–MS chromatograms also displayed the presence of metabolites characteristic of an alteration in the metabolic pathway of phenylalanine (Appendix A) (Table 1).

### 2.11. Holocarboxylase Synthetase Deficiency

In this study, the holocarboxylase synthase deficiency spectrum was described for the first time. The ^1^H-NMR spectrum of the urine sample obtained from the patient with holocarboxylase synthetase deficiency showed changes in the entire spectral region between 1.0 and 6.0 ppm when compared to the spectrum of healthy individuals. These changes were characterized by the presence of propionylglycine, 2-methyl-3-hydroxybutyric acid, acetoacetic acid, tiglylglycine, and 3-methylcrotonylglycine (Table 1) (Appendix A). On the other hand, GC–MS analysis demonstrated the occurrence of 3-hydroxypropionic acid, 3-hydroxyisovaleric acid, and methylcitric acid, in addition to propionylglycine (Appendix A) (Table 1).

### 2.12. Multivariate Statistical Analysis (MVA)

Principal component analysis (PCA) clustered the samples according to their chemical groups (Figure 3A,B). The scores plot of the PCA model (Figure 3A) distributed samples from healthy individuals in the first (upper left) and third (upper right) quadrants, along with a few IEM disorders that included maple syrup urine disease, isovaleric acidemia, glutaric acidemia type I, and 3-methylglutaconic acidemia. Meanwhile, the rest of the samples from patients with IEM disorders were separated and predominantly clustered in the second (lower left) quadrant. The loadings plot (Figure 3B) shows the type of chemical groups that were responsible for the clustering. For instance, samples from patients with IEM disorders clustering in the second quadrant mainly consisted of lipids, ketones, and aromatics. Aldehydes, purines, and amines were detected in urine samples from healthy individuals, which did not occur in the urine samples of patients with IEM disorders. However, under univariate scaling, the PCA model afforded goodness of fit (R2) and goodness of prediction (Q2) values of only 0.384 and 0.026, respectively. The quite low validation metrics were due to the occurrence of two outliers: 71MSUD (female with no record of age) and 61 β-oxidation defect (female 36 months old). Despite low R2 and Q2 values, the plots were used to visualize the type of chemical groups occurring in the urine samples of patients with IEM disorders in comparison to healthy individuals.

Additionally, least-squares discriminant analysis with orthogonal correction (OPLS-DA) (Figure 3C,D) was performed between the affected and control groups. The OPLS-DA scores plot showed distinct separation between the two classes (Figure 3C). The OPLS-DA model with Pareto scaling afforded a goodness of fit (R2) of 0.802 and a predictive ability (Q2) value of 0.714. The loadings S-plot (Figure 3D) showed a linear correlation on the increase of the occurrence of lipids, ketones, sugars, and aromatics in affected individuals; while the concentrations of purines, amines, and aldehydes significantly decreased or loss. The coefficient plot (Figure 4) indicates the predictive components unique to each group, as well as the main discriminatory metabolite for the studied disorders. However, some predictive features remained unidentified. Permutation test at n = 100 presented a Q2Y intercept of −0.238, which indicated the validity of the model [18].

In comparison to the NMR spectral dataset, the PCA of the GC–MS data provided more distinct clusters between samples taken from healthy and IEM-affected individuals as shown by the scores plot in Figure 5A that shows samples from healthy individuals in the right quadrant and samples with a few IEM disorders in the left quadrant, except for one individual with lactic aciduria (LA). The loadings plot (Figure 5B) indicates the occurrence of urea, palmitic acid (PalmAc), glutaric acid (GluAc), and hippuric acid (HPA) in the urine samples of both LA-affected and healthy individuals, though of lower quantities in urine samples of patients with LA. The best fit PCA model was obtained with Pareto scaling and log transformation that afforded goodness of fit (R2) and goodness of prediction (Q2) values of only 0.71 and 0.33, respectively. The low predictability score was due to the dispersion of IEM samples.

Similarly, OPLS-DA (Figure 5C,D) was performed between the affected and control groups. The OPLS-DA scores plot presented an even more distinct separation between the two classes (Figure 5C). With Pareto scaling, the OPLS-DA model had a goodness of fit (R2) of 0.935 and predictive ability (Q2) value of 0.825. The loadings plot (Figure 5D) defined the discriminating features such as 3-hydroxy-glutaric acid, 2-methyl-3- ketovaleric acid, 3-ketovaleric acid, 3-hydroxy-propionic acid, and 3-hydroxy-isovaleric acid for the IEM disorders of glutaric acidemia type I, holocarboxylase synthetase deficiency, phenylalanine metabolism, and propionic acidemia, respectively. 

## 3. Discussion

^1^H-NMR spectroscopy offers a complete metabolic profile by detecting different types of known or unknown metabolites, in a non-selective manner, in samples that do not need pre-treatment, unlike techniques such as GC–MS and HPLC–MS [10,19]. These chromatographic techniques coupled to mass spectrometry constitute targeted metabolomics, which are focused on analysing chemically related metabolites. Therefore, specific treatments are necessary to obtain the adequate separation and ionization of specific type of compounds, such as the liquid–liquid organic extraction steps and sample derivatization required for the GC–MS analysis of urine samples [20]. 

For the last 30 years, efforts have been made to evaluate the utility of urinary ^1^H-NMR spectroscopic profiles for detecting a wide range of smIEMs in a single assay. In fact, based on the biochemical complexity of urine samples, most diagnostic evidence has been obtained using high-resolution spectrometers of >500 MHz, which have proven to be useful for discriminating potentially pathological samples [12,13,14,15,16,17,21]. Indeed, this technology has recently been applied to newborn screening scenarios through quantitative and multivariate analyses of ^1^H-NMR spectral data [12,13]. However, compared to GC–MS, ^1^H-NMR instrumentation is expensive, so acquiring and supporting a high-resolution spectrometer might be unaffordable in some contexts [22]. Therefore, in this work, we evaluated the utility of ^1^H-NMR with the sensitivity expected for a field strength corresponding to 400 MHz that is of lower resolution than that used previously. The results demonstrated the feasibility of clearly identifying characteristic biochemical profiles for nine different smIEM disorders that were comparable and complementary to those obtained by GC–MS analysis, as previously reported in the literature [13,14,15,16,21,23,24]. 

In general, the analysed samples revealed different biochemical abnormalities using both techniques. However, some alterations implicated non-specific metabolites associated with various clinical conditions for patients who were symptomatic during the time of sampling. Some non-specific metabolites detected by both techniques were related to ketotic states such as 3-hydroxybutyric acid observed in patients with isovaleric aciduria and MSUD, as well lactic acid observed in patients with isovaleric aciduria (Appendix A) [25]. In addition, as detected by GC–MS, MSUD patients presented the high excretion of 4-hydroxy-phenylacetic acid, 4-hydroxy-phenylactic acid, and N-acetyl-tyrosine (N-acetyl-tyr) due to liver impairment (Appendix A) [25].

In comparison to the ^1^H-NMR spectral data of urine samples from healthy individuals (Figure 1 and Table 2), samples from affected individuals exhibited major changes in chemical shifts in the up-field region between 0 and 5 ppm corresponding to the occurrence of lipids, ketones, and carbohydrates. These changes in chemical shifts could have been correlated to organic acidurias, which are characterized by alterations in intermediary metabolism that lead to the excretion of certain organic acids [26,27]. Such alterations in metabolic profiles were also statistically validated by the MVA of the NMR spectral data (Figure 3 and Figure 4). Though the ^1^H-NMR spectral data of most analysed pathologies resembled the diagnostic profile obtained by GC–MS, it is notable that some metabolites were only detected by ^1^H-NMR (Table 1 and Figure 4). These metabolites included propionic acid for PA, isovalerylalanine for IVA, hexanoyl/octanoylcarnitine for β-oxidation defects, acetic acid and glucose for LA, N-acetyl-phenylalanine for PHE, and 2-methyl-3-hydroxybutyric acid for HSD. 

Qualitative results were further confirmed by multivariate analyses. An OPLS-DA regression coefficient plot (Figure 4) was employed to assess the strength and validity of the emergence of respective ^1^H-NMR peak features between two variables (healthy vs. IEM-affected individuals). Thus, the correlation coefficient (Coeff_cs_) value indicated how strongly a feature was correlated to each of the respective variables. In this case, the occurrence of a positively correlating feature with the incidence of an smIEM could be classified as a significant resonance peak used to define a predictive metabolite or biomarker. For further assessment, the significance of a predictive component for an smIEM disorder was validated by looking into the *p*-values (*p* < 0.05), false-discovery rates (FDR < 0.05), and fold-change ratios (FC) of the peaks (Table 3). The metabolites considered to pass the validation with Coeff_cs_ ≥ 0.02 that were exclusively detected by ^1^H-NMR included glucose (Coeff_cs_ = 0.020; *p* = 0.0053; FDR = 0.007; FC = 0.38), 2-methyl-3-hydroxybutyric acid (Coeff_cs_ = 0.021; *p* = 0.0006; FDR = 0.003; FC = 0.32), and N-acetyl-phenylalanine (Coeff_cs_ = 0.029; *p* = 0.0153; FDR = 0.011; FC = 0.50). The urinary excretion of the latter compound has been specifically associated with defective phenylalanine metabolism [28]. However, the clinical relevance of the urinary excretion of glucose and 2-methyl-3-hydroxybutyric acid requires further validation since such metabolites may also be affected by other conditions such as ketosis and renal function [25,28,29]. Thus, it is important to consider that our qualitative analysis may have been influenced by the low number of samples analysed per condition, especially considering that the affected individuals included in this study were symptomatic patients that could have presented clinical complications and comorbidities. This is important, particularly considering the case of propionic acid, which is not pathognomonic of propionic academia; in fact, this can observed due to bacterial contamination in healthy subjects, although this was not the case in our sample [25].

Conjugates of propionic acid in propionic acidemia and isovalerylalanine in acute episodes of isovaleric acidemia were also assessable by ^1^H-NMR that showed characteristic metabolites for each pathology (Appendix A) [28,30,31]. The ^1^H-NMR resonance for propionic acid was quite weak, only affording a correlation coefficient of 0.007, which indicated a very low concentration of the metabolite in the urine samples. However, there was a 72% (SD ± 0.70) increase of the metabolites in affected individuals. Despite a significant false-discovery rate of 0.028, the *p*-value of 0.18 could only achieve 82% confidence. On the other hand, 3-hydroxypropionic acid had a lower correlation coefficient of 0.0017 and only a 26% (SD ± 0.46) increase in affected individuals but was significant with a *p*-value of 0.013 and an FDR of 0.012. As propionic acid could only be detected by ^1^H-NMR, it seems to be a less relevant diagnostic marker than 3-hydroxypropionic acid. The analysis of isovalerylalanine (Coeff_cs_ = 0.002; *p* = 0.29; FDR = 0.033; FC = 0.81) was performed on a more diluted sample obtained from a patient during an acute episode of isovaleric acidemia exhibiting an 81% (SD ± 0.89) increase of isovalerylalanine and a relatively low but significant false-discovery rate. However, the *p*-value was quite high, resulting in a low confidence of 71%. Isovalerylalanine, which was specifically qualitatively observed in isovaleric acidemia, has been previously reported a highly clinically relevant metabolite [31]. Qualitative analysis exhibited differences according to the IEM’s clinical states and treatment, demonstrating results comparable with those described earlier in the literature [30]. These results suggested that the technique should be further exploited for the identification of different disease states in real time to track the progression of the disease or treatment as the method reaches its limit of detection. 

In this study, ^1^H-NMR displayed limited accuracy and resolution compared to the GC–MS analysis of metabolites emerging in 3-methylglutaconic acidemia (Appendix A) and glutaric acidemia type I (Appendix A). The ^1^H-NMR signals for the pathological metabolites were either weak or broad, rendering the respective essential biomarkers, 3-methylglutaconic and 3-hydroxyglutaric acids, difficult to interpret [32]. As reported earlier, 3-methylglutaconic acid should exhibit six signals at 1.99 (d), 3.65 (d), and 5.96 (m) ppm for the cis configuration and 2.14 (d), 3.28 (d), and 5.85 (m) ppm for the trans congener [14]. The isoforms were reported to be present either at a 2:1 cis:trans ratio in the urine of a patient with 3-methylglutaconic type I acidemia or a 1:1 cis:trans ratio in the urine of a patient with type IV acidemia [32]. To differentiate the occurrence of the two isomers, in addition to adjusting the pH of the samples to either pH 2.5 or 9, further 2D NMR measurements such as ^1^H-^1^H COSY and ^1^H-^13^C HSQC would be necessary [14,32]. However, in the urine samples of patients with 3-methylglutaconic acidemia examined in this study via ^1^H-NMR (Appendix A), only the presence of 3-methylglutaric acid (Coeff_cs_ = 0.009; *p* = 0.043; FDR = 0.020; FC = 0.58) was detected, which was indicated by a doublet at 1.139 ppm. The detection of 3-methylglutaric acid by ^1^H-NMR was significant with FDR and *p*-values < 0.05. Despite a correlation coefficient of only 0.009, a 58% (SD ± 0.86) increase of the metabolite was observed in affected patients. For the urine samples of individuals with glutaric acidemia type I, only the presence of glutaric acid (Coeff_cs_ = 0.0075; *p* = 0.071; FDR = 0.020; FC = 0.59) was perceivable. The occurrence of both biomarker metabolites, 3-methylglutaconic and 3-hydroxyglutaric acid, in the urine samples or IEM-affected individuals was also confirmed by GC–MS analysis (Appendix A).

^1^H-NMR profiles of samples from β-oxidation-defect-affected individuals (Appendix A) allowed for the identification of convergent signals for hexanoyl- (C6) and octanoyl-carnitine (C8). The MVA of the spectral data afforded a relatively good correlation coefficient for the carnitine resonances with a magnitude of 0.020, a significantly low false-discovery rate, and an 80% (SD ± 0.58) increase in the concentration of the carnitine metabolites in β-oxidation-defect-affected individuals. However, the high *p*-value resulted in only 85% confidence and was therefore not significant. This may also be deduced from the low number of tested individuals. However, these findings are still of great importance. Though GC–MS (Appendix A) enabled the reliable identification of the metabolites with available online databases, confirmatory β-oxidation defect diagnosis has always been based on the MS/MS analysis of acylcarnitine [33,34,35,36]. Though ^1^H-NMR has offered the possibility to analyse a wider spectrum of metabolites, it is not the appropriate technique for discriminating different carnitine esters, such as hexanoyl- and octanoyl-carnitine due to the overlapping resonances for protons on the alkyl chain. Moreover, the specificity of these metabolites is increased when separately measured in plasma or serum. Most authors have questioned the clinical utility of acylcarnitine evaluation in urine, suggesting that urinary excretion greatly varies among different disorders [37,38]. For instance, some studies have reported the occurrence of false-positive and false-negative results caused by the high variation of results, the normal presence of some acylcarnitine esters in urine of healthy controls, and the potential interference of medication and dietary artifacts [39,40]. Despite this, it would be interesting to further analyse the potential clinical utility of our results, especially when considering using ^1^H-NMR in an initial global biochemical approximation to direct further biochemical confirmatory studies.

Massive lactic aciduria was detected by GC–MS, which was analogous to the ^1^H-NMR results (Appendix A) showing an 81% (SD ± 0.71) increase in intensity of lactic acid signals (Coeff_cs_ = 0.0021; *p* = 0.31; FDR = 0.037; FC = 0.81). Although the increase of lactic acid was remarkably observed in this affected individual, it was not properly manifested by its relatively lower correlation coefficient of 0.0021 with a significant false-discovery rate; the *p*-value was >0.05 at only 70% confidence. These results are in line with the fact that elevations of lactic acid have been described for different smIEM disorders related to primary causes of lactic aciduria such as pyruvate dehydrogenase deficiency, pyruvate carboxylase deficiency, tricarboxylic acid cycle (TCA), and respiratory chain disorders, as well as other causes of secondary lactic acidosis, thus making it a very unspecific biomarker—particularly in urine samples [41,42,43]. On the other hand, with ^1^H-NMR analysis for lactic aciduria, the detection of glucose was found to be a better diagnostic marker. Glucose showed a fold-change ratio of 38% affording a significant *p*-value and false-discovery rate of <0.01.

For MSUD-affected patients, ^1^H-NMR and GC–MS profiles of their urine samples (Table 1 and Appendix A) were found to coincide with several metabolites, particularly the detection of alloisoleucine and 2-oxoisocaproic acid (also known as ketoleucine), which have been described earlier in all forms of MSUD [44]. Ketoleucine is an aberrant metabolite resulting from the incomplete breakdown of branched-chain amino acids. Ketoleucine blocks the respiratory chain, thereby compromising brain energy metabolism [28]. Similarly, elevations of lactic acid have also been detected, maybe due to the accumulation of α-keto acids that reduce the activity of the Krebs cycle and consequently increasing anaerobic glycolysis, leading to a possible alteration of energy metabolism in the brain as previously observed in a mouse model [45]. From the MVA results, significant linear increases of ketones, phenolics, and aromatics were also observed in samples acquired from affected individuals (Figure 3D). Both alloisoleucine (*p* = 0.16; FDR = 0.028; FC = 0.65) and ketoleucine (*p* = 0.075; FDR = 0.022; FC = 0.65) presented positive correlation coefficient values of 0.0170 and 0.0015, respectively, though with a 65% (SD ± 0.72 and 0.53, respectively) increase in relative concentration of the metabolites in affected individuals with MSUD. False-discovery rates were significantly low. However, as reflected by the low magnitude of correlation coefficients due to low number of samples examined and the use of very diluted samples, we reported high *p*-values at 84% and 92.5% of confidence, respectively.

The ^1^H-NMR spectral data of samples obtained from a patient with defective phenylalanine metabolism exhibited an increase of aromatic proton signals that led to the elucidation of 3-phenyllactic acid and N-acetylphenylalanine (Appendix A). The GC–MS profile (Appendix A) exhibited the occurrence of 2-hydroxyphenyl acetic acid, phenylpyruvic acid, 4-hydroxyphenyl pyruvic acid, and 4-hydroxyphenyl lactic acid in addition to 3-phenyllactic acid, while the detection of N-acetylphenylalanine was exclusive to ^1^H-NMR analysis. The contrast between the two analytical methods could be explained by their differences in the sensitivity and detection capability of certain metabolites [9]. In any case, although the urinary excretion profiles are suggestive, amino acid quantification in plasma will be needed to confirm the diagnosis and to classify the type of hyperphenylalaninemia [46,47]. Both 3-phenyllactic acid (Coeff_cs_ = 0.024; *p* = 0.033; FDR = 0.019; FC = 0.67) and N-acetylphenylalanine (Coeff_cs_ = 0.029; *p* = 0.0153; FDR = 0.011; FC = 0.50) afforded significant statistical validation metrics with measurable increases of 67% (SD ± 0.76) and 50% (SD ± 0.72), respectively, of the metabolites in affected individuals, which strongly signifies that the compounds could serve as good diagnostic biomarkers for phenylalanine metabolism disorders. 

This work has also provided an initial characterization of the ^1^H-NMR profile (Appendix A) for holocarboxylase synthetase deficiency. The identification of the metabolites from the spectral data was based on the chemical shifts reported earlier in the Human Metabolome Database (https://hmdb.ca, accessed on 1 January 2017–1 December 2017) [28] and Handbook of ^1^H-NMR Spectroscopy in Inborn Errors of Metabolism [48]. In this study, the simultaneous occurrence of both the highly specific markers propionylglycine (Coeff_cs_ = 0.011; *p* = 0.012; FDR = 0.014; FC = 0.37) and methylcrotonylglycine (Coeff_cs_ = 0.021; *p* = 0.22; FDR = 0.027; FC = 0.77) was identified. Though propionylglycine only afforded a fold change of 37% (SD ± 0.46) in affected patients and a lower correlation coefficient than methylcrotonylglycine, its occurrence was significant with a *p*-value and FDR < 0.05. On the other hand, methylcrotonylglycine displayed a relatively higher correlation coefficient and a 77% (SD ± 0.65) increase of the metabolite in the urine samples of affected individuals, thus indicating the relatively good concentration of the sample used in the analysis. To date, there has been no reports in the literature that have described the ^1^H-NMR profile of holocarboxylase synthetase deficiency. However, studies have reported ^1^H-NMR profiles of biotinidase deficiency, a disorder that is biochemically related to holocarboxylase synthetase deficiency [13,19,31,49]. Both enzymes (biotinidase and holocarboxylase synthase) are involved in the biotin cycle required for precise holocarboxylase formation, while deficiencies of both enzymes are genetic causes of multiple carboxylase deficiency [50]. Holocarboxylase is a coenzyme and an active form of human carboxylases that involves apoenzymes coupling to biotin. Though some authors have suggested that biochemical profiles of both deficiencies may be similar, ^1^H-NMR profiles reported for biotinidase deficiency through the detection of 3-hydroxyisovaleric acid, methylchrotonylglycine, and lactic acid, with the last two only present in some samples [13,19,49]. Earlier studies have reported that the urinary organic acid profile obtained by GC–MS showed elevated concentrations, mainly of 3-hydroxyisovaleric and methylchrotonylglycine [50,51,52]. The occurrence of 3-hydroxyisovaleric in holocarboxylase synthetase disorders was established as a discriminating feature by OPLS-DA of the GC–MS dataset. Here, we report a wider ^1^H-NMR profile in the analysed sample showing propionylglycine, methylchrotonylglycine, tiglylglycine, and 2-methyl-3-hydroxy-butyric acid, which was consistent with the characteristic biochemical pattern observed by GC–MS. In fact, the observed profiles were not only compatible with the diagnosis of multiple carboxylase deficiency but also resembled the biochemical findings reported in the literature for holocarboxylase synthase deficiency [53,54,55,56,57]. 

Although smIEMs are considered rare diseases, their incidence could collectively comprise around 1:2000 individuals, becoming an important cause of infant morbimortality and therefore an important factor in public health [9,58]. The early diagnosis of smIEM is crucial to getting proper and early treatment so that not only acute episodes are controlled but also long-term complications could be avoided [13]. Our results demonstrated the potential utility of an ^1^H-NMR with a field strength of 400 MHz as a diagnostic tool reinforcing the idea that ^1^H-NMR testing can contribute to early detection, which would enable early therapeutic intervention. However, the results presented here are preliminary and require further analysis, including higher numbers of patients, diseases, and analyses that consider the possible influences of age and diet, among other factors. The data presented here show that the proposed technique allows for the identification of specific pathological profiles; even different metabolic states could be distinguished in the case of isovaleric aciduria. However, sensitivity did vary amongst the different evaluated diseases. Our findings point out the potential of the technique as a screening test considering that its analysis is faster due to shorter preparation and acquisition times, less sample requirements than GC–MS and HPLC, and the allowance for the evaluation of amino acids, organic acids, carnitines, and acylglycines in the same sample. In this study, by employing ^1^H-NMR as a screening tool, seven metabolites were found to be statistically significant (*p* < 0.05) for classification as potential diagnostic markers for the indication of smIEM disorders. Though in most cases, the biochemical profile would need further confirmation, the presented analyses of the profiles proved to be useful for directing further confirmatory tests evidencing differences among aminoacidopathies, organic acidurias, and β-oxidation defects. Moreover, the detected abnormalities might help to initiate some therapeutic interventions and focus the diagnostic approach to confirm a specific Inborn Error of Metabolism.

## 4. Materials and Methods

### 4.1. Subjects

The sample population enrolled in this study consisted of 53 individuals: 36 non-affected (control group) and 17 affected by smIEM. The age of the patients oscillated between 0.2 and 168 months (6 days to 14 years old). Affected individuals were diagnosed based on symptomatology associated with IEM, which were biochemically confirmed by GC–MS (Appendix A). The classification of the patients was established as follows (Table 4): 2 patients with propionic acidemia (PA), 3 patients with isovaleric acidemia (IVA), 2 patients with 3-methylglutaconic acidemia (MGA), 4 patients with glutaric acidemia Type I (GATI), 1 patient with β-oxidation defect, 1 patient with holocarboxylase synthetase deficiency (HSD), 1 patient without specific diagnosis whose urinary GC–MS profile demonstrated massive lactic aciduria (LA), 2 patients with Maple Syrup Urine Disease (MSUD), and 1 patient with a defect in phenylalanine metabolism (PHE).

The control group was between 0 and 36 months old. Healthy individuals had normal organic acid profiles and revealed no symptoms associated with smIEM. There were no food supplements in their diet. All individuals were drug free.

Urine samples were collected after signing a parental acceptance of informed consent. Random volumes of urine sample between approximately 5 and 30 mL were collected after spontaneous voiding. Samples were coded to protect the confidentiality of individuals. All samples were stored at −20 °C until processing. Table 4 shows the characteristics of the used samples. All pathological samples were initially analysed by GC–MS and HPLC and deposited at the Inborn Errors of Metabolism Institute sample bank.

### 4.2. GC–MS 

In a 10 mL test tube, 2 mL of ethyl acetate were added to a 2 mL urine sample saturated with NaCl, 100 µL of 0.1% phenylbutyric acid as an internal standard, and 100 µL of 6N HCL. The mixture was vigorously stirred and centrifuged by 3 min at 3600 rpm. After extracting the organic phase, a second liquid–liquid extraction was carried out with 2 mL of diethyl ether. The two organic phases were combined and evaporated to dryness with nitrogen. Extracted organic acids were methylsylated with N,O,-bis- (trimethylsilyl) trifluoroacetamide (BSTFA), and subsequent separation by gas chromatography was carried out by gas chromatography (HP 6890. Provider by Hewlett-Packard GmbH, Waldbronn Analytical Division. Heweltt-Packard Straße 8. 76333 Waldronn, Germany) using an HP1 polymethyl-siloxane column (0.200 mm × 12 m × 0.33 µm.) as the stationary phase and helium as the carrier gas. Separation was achieved by setting a gradual temperature increase from 80 to 280 °C in a run of 35 min. Dereplication was accomplished via electronic impact mass spectrometry at 230 MeV at 230 °C using a mass selective detector (HP 5973. Provider by Hewlett-Packard GmbH, Waldbronn Analytical Division. Heweltt-Packard Straße 8. 76333 Waldronn, Germany) [59,60].

### 4.3. GC–MS Data Analysis

The qualitative interpretation of the chromatogram was performed by a medical laboratory scientist with experience in the laboratory diagnosis of organic acidurias. Chromatographic peaks were manually selected, and identification was performed based on the results of the mass spectrum comparison against ORG_ACID, ORGACIDS, and NIST17 libraries for organic compounds [61]. GC–MS profile interpretation relied on the identification of the increased concentration of normally occurring metabolites and/or the presence of abnormal metabolites, as reported in the literature.

### 4.4. ^1^H-NMR Sample Preparation 

Urine samples were prepared following a modified procedure of a previously described protocol [12,62]. Briefly, sample thawing was allowed for up to 60 min and centrifuged at 12,000 rpm for 10 min. Then, 540 µL of urine (at room temperature) were mixed with 180 µL of a 1.5 M phosphate buffer (pH 7) containing 0.5 mM 3-(trimethylsilyl) propionic 2,2,3,3-d4 acid sodium (TSP-d_4_) as the internal standard. Furthermore, 5 mM ^1^H-NMR tubes were used to record the spectra. The signal of TSP was used as the chemical shift reference at 0.0 ppm.

### 4.5. ^1^H-NMR Data Acquisition and Processing

All ^1^H-NMR spectra were recorded at 296 °K on Bruker Advance(provide by Bruker BioSpin AG. Idustiestrasse 26. CH-8117 Falladen. Denmark) with a field strength of 400 MHz ^1^H-NMR spectrometer using a pulse sequence, NOESYPR1D, with the pre-saturation of the water peak. Each spectrum was accumulated with 64 scans, a delay time of 4 s, and an acquisition time of 3.41 min with 32 Kb; the pulse attenuation for pre-saturation was at 52.18 dB, with a pulse of 12.30 us. Spectra were phased in MNOVA^®^ version 10 [63]. The analysed signals were located between 1 and 9 ppm; water, urea, and TSP-d_4_ signals were excluded. iCOshift toolbox 3.1.1 implemented for MATLAB^®^ version R2016b was used to prepare the data for statistical analysis [64]. 

### 4.6. Qualitative Profile Identification 

^1^H spectral resonances for the metabolites were assigned while considering the chemical shift values reported in the human metabolome (HMDB) and previous studies [19,65,66]. The Biological Magnetic Resonance Data Bank (BMRB) databases [67] and Resurrecting and Research tool. ^1^H-NMR Spectra were processed on-line (https://www.nmrdb.org/, accessed on 1 January 2017–1 December 2017), along with information available in the literature [14,16,17,62,68,69,70,71,72,73,74,75,76,77,78,79].

### 4.7. Multivariate Analysis (MVA)

The ^1^H-NMR spectral dataset was further analysed using SIMCA V17 (Umetrics, Umeå, Sweden). An unsupervised principal components analysis (PCA) was used to reduce the number of dimensions for further multivariate statistical analyses. Additionally, a supervised multivariate analysis (orthogonal partial least squares discriminant analysis; OPLS-DA) was also performed to predict the discriminatory features for each assigned class [63,80,81]. Validation metrics, including the goodness of fit and prediction, and permutation tests on PLS models were used. For the detection of biomarker metabolites, validation metrics such as *p*-values, fold changes, and false-discovery rate (FDR) were used to evaluate the significance of the respective metabolites as potential diagnostic biomarkers. In calculating the FDR, equations from Benjamini–Hochberg [82], Holm’s and Hochberg [83], Dunn–Sidàk [84], and Benjamini–Yekutieli [84,85,86] were utilised to cross-validate the significance of the results from which the mean average was used.

The GCMS dataset was pre-processed using Mzmine 2.53 (http://mzmine.github.io/, accessed on 1 January 2017–1 December 2017) following the protocol from https://ccms-ucsd.github.io/GNPSDocumentation/gc-ms-data-processing-MZmine2/, accessed on 1 January 2017–1 December 2017. The processed dataset was subjected to MVA using SIMCA as used for the ^1^H-NMR spectral dataset.

## 5. Conclusions

The combination of GC–MS and ^1^H-NMR as metabolomic approach used in this study allows one to obtain a more holistic view of smIEM disorders such as phenylalanine metabolism; isovaleric, propionic, 3-methylglutaconic, and glutaric type I acidemia; and deficiencies in medium chain acyl-coenzyme and holocarboxylase synthase. In this work, we highlight the use of ^1^H-NMR as a screening tool for small molecules correlated to Inborn Errors of Metabolism considering that it is a rapid, accurate, and effective method to detect a variety of inherited metabolic disorders using only 540 µL of urine that also requires low processing and interpretation times. Our findings demonstrated that by using low-resolution equipment (400 MHz), it was possible to detect abnormalities in ^1^H-NMR spectra in samples from patients with different organic acidemias, β-oxidation defects, and aminoacidopathies. These data, in combination with GC–MS, allowed for a more comprehensive analysis of the of the dysregulated metabolic pathways associated with smIEM. Additionally, to the best of our knowledge, this constitutes the first description of the ^1^H-NMR spectral profile of holocarboxylase synthase deficiency. The results presented here are encouraging for the implementation of ^1^H-NMR as a complementary method for screening and monitoring IEM disorders. The ^1^H-NMR spectral data afforded indicative profiles that could lead to further diagnostic studies and the implementation of early simple lifestyle interventions (e.g., avoidance of fasting or nutritional restrictions) that may save the lives of patients and avoid irreversible clinical consequences.

## Figures and Tables

**Figure 1 metabolites-11-00891-f001:**
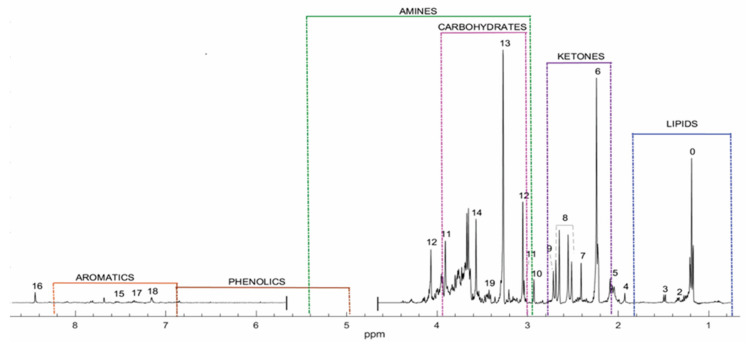
Average spectrum at 400 MHz of urine samples obtained from healthy individuals. Chemical families were classified into lipids, ketones, carbohydrates, amines, phenolics, aromatics, and aldehydes.

**Figure 2 metabolites-11-00891-f002:**
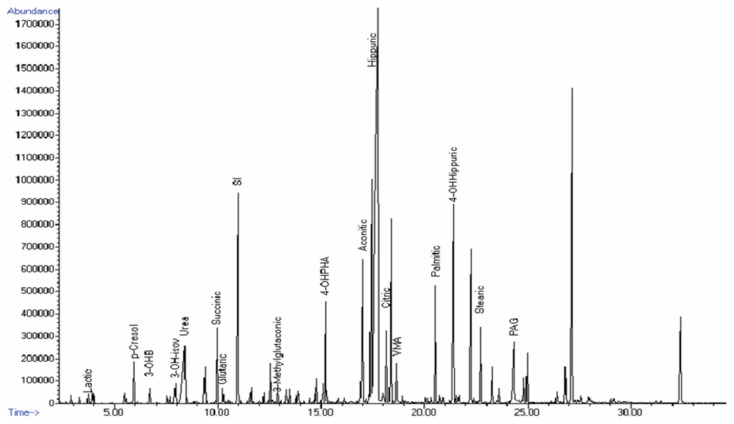
GC–MS profile chromatogram from healthy individuals. Peaks detected for lactic acid, p-cresol, 3-hydroxybutyric acid (3-OHB), 3-hydroxyisovaleric acid (3-OH-isov), urea, succinic acid, glutaric acid, internal standard (SI), 3-methylglutaconic acid, 4-hydroxyphenylacetic acid (4-OHPHA), aconitic acid, hippuric acid, citric acid, vanillylmandelic acid (VMA), palmitic acid, 4-hydroxy-hippuric acid, stearic acid, and phenylacetyl glutamine (PAG).

**Figure 3 metabolites-11-00891-f003:**
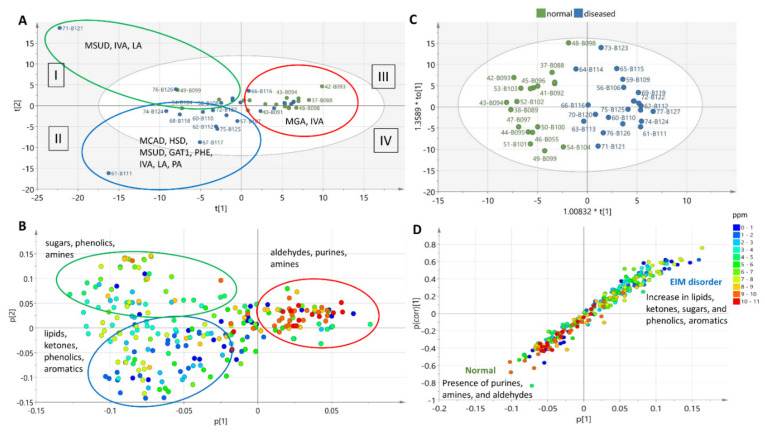
Multivariate analysis of ^1^H-NMR spectral data of healthy samples vs. those with diagnosed IEM disorders. (**A**) PCA scores plot, (**B**) PCA loadings plot, (**C**) OPLS-DA scores plot, and (**D**) OPLS-DA loadings S-plot. Legend: propionic acidemia (PA), isovaleric acidemia (IVA), medium chain acyl-CoA dehydrogenase deficiency (MCAD), 3-methylglutaconic acidemia (MGA), holocarboxylase synthetase deficiency (HSD), glutaric acidemia type I (GAT1), lactic aciduria (LA), maple syrup urine disease (MSUD), and disorders of phenylalanine metabolism (PHE).

**Figure 4 metabolites-11-00891-f004:**
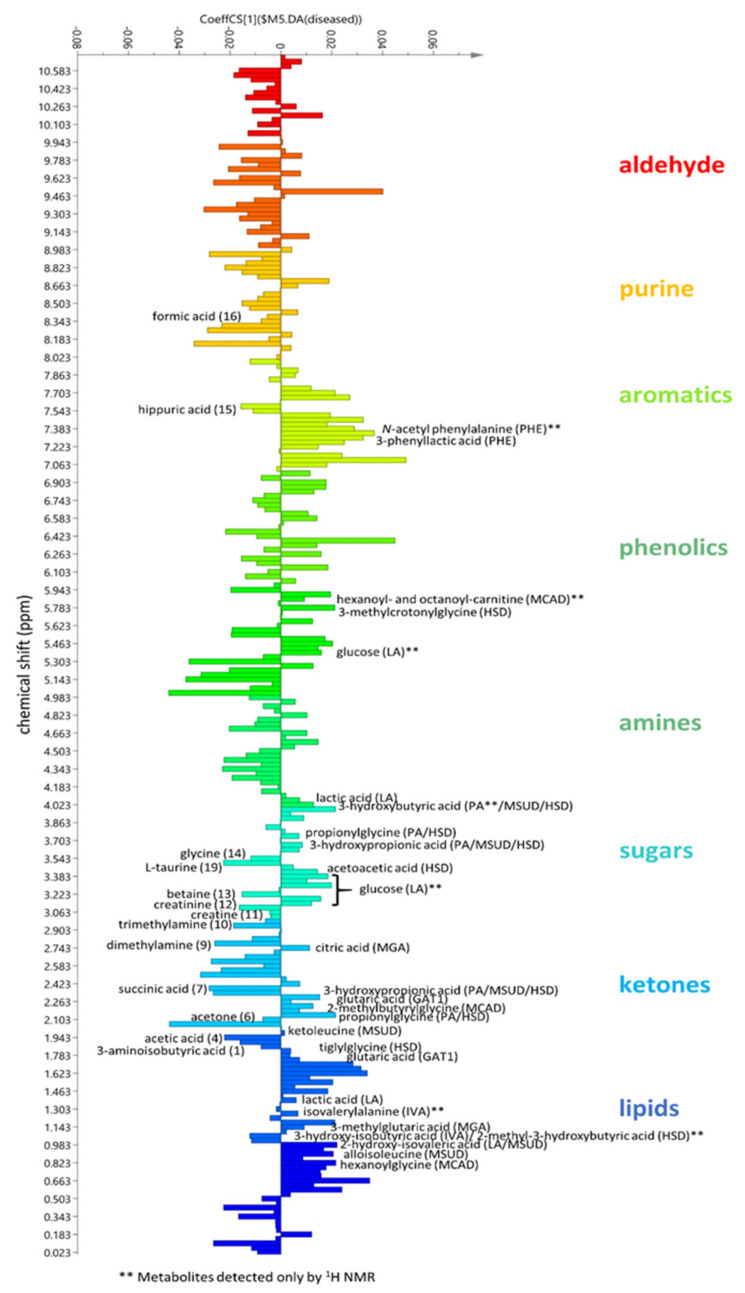
OPLS-DA regression coefficient plot of ^1^H-NMR spectra indicating the relative occurrence of respective metabolites between healthy and affected individuals, as listed in Table 1 and Table 2. Unlabelled peaks are unidentified features.

**Figure 5 metabolites-11-00891-f005:**
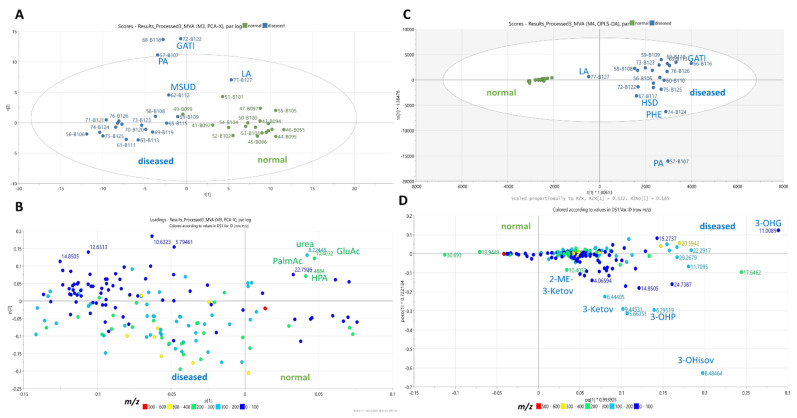
Multivariate analysis of GC–MS data of healthy samples vs. samples with diagnosed IEM disorders. (**A**) PCA scores plot, (**B**) PCA loadings plot, (**C**) OPLS-DA scores plot, and (**D**) OPLS-DA loadings plot. Legend: propionic acidemia (PA), holocarboxylase synthetase deficiency (HSD), glutaric acidemia type I (GAT1), lactic aciduria (LA), maple syrup urine disease (MSUD), and disorders of phenylalanine metabolism (PHE). Labelled features with their retention time correspond to identified metabolites from the in-house in line database such as urea, palmitic acid (PalmAc), glutaric acid (GluAc), hippuric acid (HPA), 3-hydroxy-propionic acid (3OHP), 3-ketovaleric acid (3-Ketov), 2-methyl-3- ketovaleric acid (2-ME-3-Ketov), 3-hydroxy-isovaleric acid (3-OHisov), and 3-hydroxy-glutaric acid (3-OHG).

**Table 1 metabolites-11-00891-t001:** Metabolites observed by GC–MS and ^1^H-NMR. Metabolites in italics were only detected by ^1^H-NMR. Representative spectra of urine samples from respective individuals with certain IEM disorders are shown in the Appendix A.

	Abnormal Metabolites Detected by GC–MS		Abnormal Metabolites Detected by ^1^H-NMR		
IEM	Metabolite	Retention Time (min)	Metabolite	ppm	Peak Feature
Propionic Aciduria	3-Hydroxypropionic Acid	6.316	*Propionic Acid*	1.038	triplet
				2.286	quartet
			3-Hydroxypropionic Acid	2.429	triplet
	3-Hydroxyvaleric Acid	8.084	Propionylglycine	1.119	triplet
	Propionylglycine	12.132		2.301	quartet
	Tiglylglycine	14.104		3.75	quartet
	Methylcitric Acid	18.513	3-Hydroxypropionic Acid	2.429	triplet
				3.787	triplet
			*3-Hydroxybutyric Acid*	4.103	multiplet
Isovaleric Aciduria	3-Hydroxyisovaleric Acid	8.145	3-Hydroxyisovaleric Acid	1.275	singlet
				2.35	singlet
	Isovalerylglycine	12.903	Isovalerylglycine	0.9332	doublet
				1.975	multiplet
				2.372	doublet
	Isovalerylglutamate	19.447	*Isovalerylalanine* ^†^	1.33	doublet
				4.12	multiplet
β-oxidation defect	Octanoic Acid (Dehydrosuberic Acid)	15.944	Butyrylglycine	2.27	triplet
			Isovalerylglycine	1.975	multiplet
	Adipic Acid	13.415	Butyrylglycine—*Suberylglycine* (convergent signals)	1.860–1.983	multiplet
	Suberic Acid	16.22			
	Tiglylglycine	14.265			
	2-Methylbutyrylglycine	13.289	2-Methylbutyrylglycine	1.19	doublet
			2-Methylbutyrylglycine	2.26	triplet
	Hexanoylglycine	15.013	*Hexanoylglycine; 2-Methylbutyrylglycine* (convergent signals)	1.983–2.119	multiplet
			Hexanoylglycine	0.8689	triplet
			Hexanoylglycine	2.24	multiplet
	Octanoylglycine	15.447	Hexanoylcarnitine—Octanoylglycine (convergent signals)	5.77–5.85	multiplet
	Isobutyrylglycine	16.006	Isobutyrylglycine	1.239	doublet
3-Methylglutaconic aciduria	3-Methylglutaric Acid	12.132	3-Methylglutaric Acid	1.139	doublet
	3-Methylglutaconic Acid	13.327			
Holocarboxylase Synthetase Deficiency	3-Hydroxypropionic Acid	6.326	Propionylglycine—2-Methyl-3-hydroxybutyric Acid	1.119	doublet
	3-Hydroxyisovaleric Acid	8.729	*2-Methyl-3-hydroxybutyric Acid*	2.09	doublet
	Propionylglycine	12.132		1.198	doublet
	Isobutyrylglycine TMS II	12.176	Acetoacetic Acid	2.28	singlet
	3-Methylcrotonylglycine		Tiglylglycine	3.456	singlet
		14.849		1.77	singlet
	Methylcitric Acid	18.466	3-Methylcrotonylglycine	5.77	multiplet
Glutaric Aciduria Type I	Glutaric Acid	11.485	Glutaric Acid	1.752	multiplet
	Glutaconic Acid	11.535		2.172	triplet
	3-Hydroxyglutaric Acid	14.366			
Phenylalanine Metabolism Disorders	2-Hydroxyphenylacetic Acid	14.415	3-Phenyllactic Acid	2.861	doublet doublet
	3-Phenyllactic Acid	14.825		7.33	multiplet
	4-Hydroxyphenylpyruvic Acid	16.482	*N-acetyl-phenylalanine*	1.928	singlet
				7.32	multiplet
	4-Hydroxyphenyllactic Acid	19.085			
Lactic Aciduria	Lactic Acid	3.671	Lactic Acid	1.33	doublet
	2-Hydroxyisobutyric Acid	5.408		4.102	quartet
	4-Hydroxyphenyllactic Acid	18.614	*Acetic Acid*	1.93	singlet
			*Glucose*	3.241	multiplet
				3.396	multiplet
				3.458	multiplet
				3.536	doublet doublet
				3.61	multiplet
				3.702	multiplet
				3.895	doublet
				4.655	doublet
				5.24	doublet
Maple Syrup Urine Disease (MSUD)	2-Hydroxyisovaleric Acid	7.375	3-Hydroxybutyric Acid	2.3	multiplet
				4.103	multiplet
			3-Methyl-2-oxovaleric Acid	1.2	doublet
	3-Methyl-2-oxovaleric Acid	8		1.42	multiplet
	3-Hydroxyisovaleric Acid	8.084	2-Hydroxyisovaleric Acid	0.92	doublet
	2-Oxoisocaproic Acid (Ketoleucine)	8.75		0.98	doublet
	Alloisoleucine	12.116	Alloisoleucine	0.94	multiplet
			Isoleucine	0.926	triplet
				1.42	multiplet
			*2-Oxoisocaproic Acid (Ketoleucine)*	0.92	doublet
				1.99	multiplet
				2.6	doublet

^†^ Present only in acute episodes.

**Table 2 metabolites-11-00891-t002:** ^1^H-NMR chemical shift values in ppm identified in NMR spectra for detected metabolites from healthy individuals.

No	COMPOUND	ᵟ^1^H (pH = 7.0)
0	Unidentified (found in the area corresponding to lipids).	1.170 (t)
1	3-Aminoisobutyric Acid	1.187–1.212 (d)
2	Lactic Acid	1.33–1.347 (d)
3	Alanine	1.475–1.499 (d)
4	Acetic Acid	1.93 (s)
5	N-Acetyl region	1.991–2.027–2.044–2.066–2.09 (s)
6	Acetone	2.24 (s)
7	Succinic Acid	2.412 (s)
8	Citric Acid	2.509–2.56–2.651–2.70 (d,d)
9	Dimethylamine (DMA)	2.717 (s)
10	Trimethylamine (TMA)	2.931 (s)
11	Creatine	3.03–3.939 (s,s)
12	Creatinine	3.05–4.071 (s,s)
13	Betaine	3.27 (s)–3.91 (s)
14	Glycine	3.57 (s)
15	Hippuric Acid	3.96–7.52–7.61–7.82 (d,t,t,d)
16	Formic Acid	8.44 (s)
17	ɑ-N-Phenylacetyl-L-glutamine	7.328–7.391 (t)
18	L-Histidine	7.156 (s)
19	L-Taurine	3.426 (t)

ᵟ^1^H: Chemical displacement (ppm), singlet (s), doublet (d), triplet (t).

**Table 3 metabolites-11-00891-t003:** Validated biomarker metabolites as potential diagnostic compounds for smIEM profiling.

smIEM Disorder	Metabolite	Coeff_cs_	%FC (SD)	FDR	*p*-Value	n
Propionic acidemia	*Propionic acid*	0.007	72% (±0.70)	0.028	0.180	17
	3-hydroxy propionic acid	0.0017	26% (±0.46)	0.012	0.013 *	22
Isovaleric acidemia	*Isovalerylalanine*	0.002	81% (±0.89)	0.033	0.290	18
3-methyl glutaconic acidemia	3-methylglutaric acid	0.009	58% (±0.86)	0.020	0.043 *	21
Glutaric acidemia Type I	Glutaric acid	0.007	59% (±0.80)	0.020	0.071	17
β-oxidation defect	*Hexanoyl/octanoyl carnitine*	0.020	80% (±0.58)	0.026	0.153	7
Lactic acidemia	*Glucose*	0.020	38% (±0.53)	0.007	0.005 **	20
	Lactic acid	0.002	81% (±0.71)	0.037	0.310	16
Maple syrup urine disease	Alloisoleucine	0.017	65% (±0.72)	0.028	0.160	17
	Ketoleucine	0.002	65% (±0.53)	0.022	0.075	18
Phenylalanine metabolism	*N*-acetyl-phenylalanine	0.029	50% (±0.72)	0.011	0.015 *	16
	3-phenyllactic acid	0.024	67% (±0.76)	0.019	0.033 *	13
Holocarboxylase synthetase deficiency	*2-methyl-3-hydroxy* *butyric acid*	0.021	32% (±0.72)	0.003	0.0006 **	22
	Propionylglycine	0.011	37% (±0.46)	0.014	0.012 *	18
	Methylcrotonylglycine	0.021	77% (±0.65)	0.027	0.220	5

* Significant at *p* < 0.05; ** Significant at *p* < 0.01; Coeff_CS_: correlation coefficient; %FC (SD): % fold change (standard deviation); FDR: false discovery rate; n: number of collected resonances; in italics are metabolites exclusively detected by ^1^H-NMR.

**Table 4 metabolites-11-00891-t004:** Characteristics of affected individuals.

No	ID	Gender	Age (Month)	Condition *	Biochemical Diagnosis	Classification
1	56SV929436PA	M	9	Follow up	Propionic aciduria	Organic acidurias
2	57PA	M	0.52	Diagnosis		
3	58IVA	F	1	Diagnosis	Isovaleric aciduria	
4	59IVA	NA	1	Diagnosis		
5	60IVA	F	168	Follow up		
6	64MGA	F	2	Diagnosis	3-Methylglutaconic aciduria **	
7	65MGA	F	0.79	Diagnosis		
8	68GATI	M	36	Follow up	Glutaric aciduria type I	
9	69GATI	F	48	Follow up		
10	70GATI	M	72	Follow up		
11	72GATI	F	72	Follow up		
12	61 β-oxidation defect;	F	36	Follow up	β-oxidation defect	Fatty Acid Oxidation Disorder
13	67HSD	F	0.72	Diagnosis	Holocarboxylase synthetase deficiency	Amino Aciduria
14	77LA	M	0.23	Diagnosis	Lactic aciduria	Mitochondrial disorder
15	62MSUD	M	24	Follow up	Maple syrup urine disease	Aminoacidopathies
16	71MSUD	F	NA	Follow up		
17	74PHE	F	144	Follow up	Disorders of phenylalanine metabolism	

Abbreviation: M: male; F: female; NA: data not available. * The condition refers to the context in which the sample was processed for either diagnosis or follow-up. ** It is unclear the specific subtype since diagnosis was based on the organic acid profile and no molecular or enzymatic testing was performed.

## Data Availability

All the information related to the study protocol and its results are available in the repository of the Pontficia Universidad Javeriana, included in the Master’s Degree work entitled: “Análisis de Metabolitos Urinarios Detectados por Resonancia Magnética Nuclear Protónica (RMN1H) en Pacientes con Errores Innatos del Metabolismo”. Pulido Ochoa NF. 2017.

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
