# Peer review of "1H-Nuclear Magnetic Resonance Analysis of Urine as Diagnostic Tool for Organic Acidemias and Aminoacidopathies"

_metabolites, 2021, doi:10.3390/metabo11120891_

Round 1
Reviewer 1 Report
The work describes organic acidemias and aminoacidopathies using NMR and GC-MS, however the GC-MS information is not found in the exploited body of work. And NMR is well discussed and is an excellent proposition to recognize these biochemical diseases.
The analysis of the data by nuclear magnetic resonance is clear, but the metabolomic analysis by GC-MS was not performed. To compare the methods. I suggest adding the statistical analysis by GCMS, including the PCA, OPLSDA, including figures in supporting material. Discuss the differences between the methodologies used.
Author Response
The authors want to thank the reviewers for their comments. The manuscript has been thoroughly revised; thus, we believe it has greatly improved. Please note that corrections are highlighted in yellow within the manuscript.
The work describes organic acidemias and aminoacidopathies using NMR and GC-MS, however the GC-MS information is not found in the exploited body of work. And NMR is well discussed and is an excellent proposition to recognize these biochemical diseases.
The analysis of the data by nuclear magnetic resonance is clear, but the metabolomic analysis by GC-MS was not performed. To compare the methods. I suggest adding the statistical analysis by GCMS, including the PCA, OPLSDA, including figures in supporting material. Discuss the differences between the methodologies used.
R//. The GC-MS multivariate analysis were included within the manuscript (lines 351-370 and Figure 5) and differences between both techniques were further explained in the initial part of the discussion (lines 376-381).
Reviewer 2 Report
1. I am alarmed by the fact that the age of the patients in the main group varies from 0 to 14 years, while in the control group it is from 0 to 3 years. This does not allow the statisticians to correctly compare groups with each other. 2. The main group includes a fairly large number of subgroups with different metabolic disorders. Subgroups include from 1 to 4 people, which also raises doubts about the results obtained and their reproducibility. I believe that the authors should enter in the title and specify according to the text of the manuscript that the results obtained are preliminary. 3. Minor remarks: figure captions are given above figures, but should be at the bottom. Significant digits and separators (periods or commas) should be checked in tables.
Author Response
The authors want to thank the reviewers for their comments. The manuscript has been thoroughly revised; thus, we believe it has greatly improved. Please note that corrections are highlighted in yellow within the manuscript.
1.I am alarmed by the fact that the age of the patients in the main group varies from 0 to 14 years, while in the control group it is from 0 to 3 years. This does not allow the statisticians to correctly compare groups with each other.
R//. Although changes in the metabolic profile have been associated to age, diet and ethnicity, in our experience the main alterations within the urinary organic acid profile of individuals occurs during the first year of life due to dietary transition during this period. Similar findings have also been reported by other authors like Boulat et al 2005 and Kumari 2015. In addition, for most pathologic cases, metabolic changes imply compounds that are not present in healthy individuals.
Despite of this, we agree with your comment and we included a mention of this limitation in the discussion of the manuscript (lines 598-600).
Boulat, O. G., Marianne. Matos, Vera. Guignard, Jean-Pierre. Bachmann, Claude. (2005). "Organic Acids in the second morning urine in a Healthy Swiss Paedriatic Population " Clinical Chemistry and Laboratory Medicine 41(12): 1642–1658.
Kumari, C. S., Ankur. Ramji,Siddharth. Shoemaker, James D. Kapoor, Seema. (2014). "Urinary Organic Acids Quantitated in a Healthy North Indian Pediatric Population." Association of Clinical Biochemists of India 30(2): 221–229.
- The main group includes a fairly large number of subgroups with different metabolic disorders. Subgroups include from 1 to 4 people, which also raises doubts about the results obtained and their reproducibility. I believe that the authors should enter in the title and specify according to the text of the manuscript that the results obtained are preliminary.
R//. We agree with your comment and therefore we state that our results are preliminary (lines 598-600).
- Minor remarks: figure captions are given above figures, but should be at the bottom. Significant digits and separators (periods or commas) should be checked in tables.
R// We apologize for the mistake, the captions were placed below figures and periods/commas were checked.
Reviewer 3 Report
Materials & Methods
Lines 124-125, the authors state that “Urine samples were collected prior to signing a parental acceptance of informed consent.” If collected prior to signing consent, there should be a follow-up statement showing that consent was obtained or why it was not needed.
Lines 145-146 – How many peaks were identified manually? The authors then mention three libraries of organic compounds. Can the comparisons not be done in an automated fashion?
Section 2.4 – Why was it necessary to allow the samples to stand at room temperature for 60 minutes? Temperature and storage conditions has been shown to impact the urine metabolite profile.
Results
Table 2 – For Isovaleric Aciduria, there are a couple of metabolites in both GC-MS and NMR; put them next to each other in the table. Same comment for Beta-Oxidation Defect.
Lines 226-227, the authors note discrepancies to detecting metabolites between the two techniques. This is not unexpected as the different analytical technologies, i.e. NMR, LC-MS, GC-MS, will be better at detecting certain classes of compounds compared to others. I don’t think that "discrepancies" is the correct way to state this.
Why is all of the analysis qualitative rather than quantitative?
Line 254, switch order of “and” and “acid” after “3-hydroxyglutaric”.
Were metabolites positively identified with authentic standards or just based upon database information?
Line 338 – The authors state that mass spectrometry detects a limited variety of compounds; I would argue that the same could be said for NMR since the compounds with high enough concentration to be measured by NMR are generally from a few major classes of analytes.
Lines 340 and on – How many hospitals/clinics are using NMR to evaluate smIEM? Isn’t a mass spectrometry-based platform typically used?
Line 565 – No quantitative analysis was done.
Delete lines 569-571
Author Response
The authors want to thank the reviewers for their comments. The manuscript has been thoroughly revised; thus, we believe it has greatly improved. Please note that corrections are highlighted in yellow within the manuscript.
Materials & Methods
Lines 124-125, the authors state that “Urine samples were collected prior to signing a parental acceptance of informed consent.” If collected prior to signing consent, there should be a follow-up statement showing that consent was obtained or why it was not needed.
R//. This was a writing mistake, samples were obtained after signing the informed consent. This was corrected within the manuscript.
Lines 145-146 – How many peaks were identified manually? The authors then mention three libraries of organic compounds. Can the comparisons not be done in an automated fashion?
R// The protocol used in our laboratory is performed by manual inspection of the chromatogram and comparison against the libraries. The mentioned lines were adjusted to clarify this point.
Section 2.4 – Why was it necessary to allow the samples to stand at room temperature for 60 minutes? Temperature and storage conditions has been shown to impact the urine metabolite profile.
R//. The time mentioned corresponded to that maximum time that allowed sample thawing and it could achieve room temperature to assure homogeneous sample processing temperature. This was clarified within the manuscript.
Results
Table 2 – For Isovaleric Aciduria, there are a couple of metabolites in both GC-MS and NMR; put them next to each other in the table. Same comment for Beta-Oxidation Defect.
R//. Similar metabolites were adjusted so similar findings were together.
Lines 226-227, the authors note discrepancies to detecting metabolites between the two techniques. This is not unexpected as the different analytical technologies, i.e. NMR, LC-MS, GC-MS, will be better at detecting certain classes of compounds compared to others. I don’t think that "discrepancies" is the correct way to state this.
R//. Thanks for the comment, we agree with the reviewer. In this sense, and in order to avoid a negative connotation we changed the word discrepancies by differences.
Why is all of the analysis qualitative rather than quantitative?
R// For diagnosis of organic acidurias evidence of abnormal metabolites qualitatively is enough for stablishing diagnosis. Therefore, our laboratory has not implemented quantitative analyses either of GC-MS and 1H-NMR.
Line 254, switch order of “and” and “acid” after “3-hydroxyglutaric”.
Delete lines 569-571
R// Corrected.
Were metabolites positively identified with authentic standards or just based upon database information?
R// Identification was based only in database information.
Line 338 – The authors state that mass spectrometry detects a limited variety of compounds; I would argue that the same could be said for NMR since the compounds with high enough concentration to be measured by NMR are generally from a few major classes of analytes.
R//. This sentence was modified in order to clarify differences between both techniques (Lines 376-381).
Lines 340 and on – How many hospitals/clinics are using NMR to evaluate smIEM? Isn’t a mass spectrometry-based platform typically used?
R//. NMR is not widely used, as you mentioned the gold standard platforms are based on mass spectrometry, however center like Özel Bahat Hospital in Turkey are currently using NMR for clinical purposes.
Line 565 – No quantitative analysis was done.
R//. We changed the sentence to “the presented analyses of the profiles…”
Round 2
Reviewer 1 Report
Many thanks to the authors for the corrections made. The work has improved substantially, and my doubts regarding the work have been resolved. Thanks for solving my doubts.Reviewer 2 Report
I have no more comments on the article, I believe that in its present form it can be recommended for publication.